# Competitive Green Supply Chain Transformation with Dynamic Capabilities—An Exploratory Case Study of Chinese Electronics Industry

Ying Ye [1,*] and Kwok Hung Lau [2]

1   Department of Management Science, School of Politics and Public Administration, Soochow University, Suzhou 215006, China
2   School of Accounting, Information Systems and Supply Chain, College of Business and Law, RMIT University, Melbourne, VIC 3000, Australia; charles.lau@rmit.edu.au
*   Correspondence: yingye@suda.edu.cn

**Abstract:** Extant studies identify different strategic approaches for businesses to integrate environment management (EM) into corporate supply chain management (SCM) to create different competitive market advantages. With increasing market uncertainties, stakeholder complexities, constraints of resources, companies nowadays are required to develop more context-specific solutions for mitigating sustainability risk and improving market performance in supply chain sustainability practices. To address the challenges faced by businesses, this study aims to explore how a company adapts green supply chain strategies for a competitive transformation. Specifically, we aim to identify different green supply chain capabilities that form hybrid choices of green solutions for a company to leverage an adaptive green shift. Through an extensive literature review, this study proposes a conceptual framework for exploration which is based on a qualitative in-depth case study in the electronics industry—one of the most polluting manufacturing sectors in China. The findings show that supply chain characteristics play a significant role in the selection of different green SCM solutions by businesses. We find different hybrid choices of strategic decisions are being made by the case company operating with high-volume and high-value supply chains respectively. High-volume supply chains adopt both operational pollution and waste control strategies and incremental green product strategies to balance cost of operational change and develop a competitive eco-efficient market expansion. In contrast, high-value supply chains adopt radical green product innovation strategies and operational waste prevention strategies to balance cost of product innovation and build operational efficiency to expand in a competitive eco-differentiation market. A mix of dynamic capabilities involved in different supply chain green transformations are also identified.

**Keywords:** green supply chain strategy; green operation and relationship capability; lean and green; incremental and radical green product innovation

## 1. Introduction

Green supply chain management (GSCM) has become an essential initiative that drives industrial value and competitive advantages, rather than a voluntary obligation in conflict with traditional corporate economy objectives [1,2]. Many studies propose different strategic approaches for businesses to integrate environment management (EM) practices into corporate supply chain management (SCM) to create different competitive market advantages [3,4]. For instance, research shows that through adoption of pollution control and prevention strategies, a company can build a measurable EM control on internal operations of production and logistics to reduce risks of violation in environmental compliance and reduce internal wastes of supply chain operation [5,6]. Other studies [7–9] also propose that, through adoption of green product strategy, businesses can develop

differentiated products and build market novelty through creation of sustainable products that can have less—or even zero—impact on the environment.

The electronics industry, commonly regarded as one of the most polluting manufacturing sectors in China, generates waste at multiple stages of its supply chain—during production and end-of-life recycling [10]. Upstream, cheap component production from suppliers leads to extensive water and chemical pollutants. Downstream, unauthorized and crude ways of recycling to extract precious metals and organic materials to be sold for economic value leads to the leaking of huge amounts of toxic chemicals, such as mercury, lead, and cadmium [11]. This is further exacerbated by the rapid growth of the electronic product market fueled by young consumers, as well as the growing desire for the latest technological products leading to shorter and shorter product lifecycles and increasing e-wastes [12]. According to the data of China Household Electrical Appliances Research Institute, 170 million units of electrical and electronic product waste were generated in 2019—of which 20.4 million units were microcomputers [13].

Despite different strategies having been proposed for companies to green their supply chains, manufacturers are facing difficulties in adopting green practices to the entire supply chain. On the one hand, there is a lack of supply infrastructure for green sourcing and production. Upstream suppliers operating at a small scale with thin profit margins are not able to change their current practices of operation easily due to resource constraints [14]. Manufacturers that are predominantly assembly-based and not involved in heavy industrial production and emission lack direct incentives to require their upstream suppliers to take pollution control measures. On the other hand, there is a lack of consumer demand for green products in the current market. Awareness of green consumption for Chinese consumers is only at the nascent stage of development. It is challenging for manufacturers to adopt green product strategies to obtain market advantage since the demand is unpredictable and upfront technological investment is high [15,16]. Thus, how companies can leverage a competitive green transformation that stimulates green demand while balancing a supply pollution control cost tradeoff is crucial for contemporary Chinese manufacturers under dynamic industrial uncertainties.

Recent research [17,18] proposes that more context-specific solutions need to be developed for mitigating sustainability risk and improving market performance in supply chain sustainability practices. Rauer and Kaufman (2015) [19] suggest that businesses competing in a changing environment nowadays should take a dynamic capability view to develop: (i) industrial and market sensing capabilities; (ii) vertical and stakeholder alignment; and (iii) technical and social resilience capabilities. These capabilities are required to overcome barriers arising from a lack of regulatory operational control, lack of external collaboration, and risk of deployment failure resulting from internal and external uncertainties. Beske et al., (2014) [17] also suggest businesses should develop bundles of capabilities and build dynamic capacities to reconfigure the resource base for long-term sustainable competitive advantage. Underpinned by the above, this study aims to answer the following research question:

How does a company adopt green supply chain strategies for a competitive transformation?

Specifically, we aim to identify: (i) what green supply chain capabilities are required for businesses to leverage different green supply chain strategies; and (ii) how a competitive strategic transformation is achieved through building dynamic green supply chain capabilities. To this end, this research adopts a qualitative case study methodology to explore the issues in the context of the emerging Chinese market. Compared to developed Western markets, GSCM in China is still at an infant stage of business management [14,20,21]. Since different conditions of market resources and institutions can influence the way that a business adopts EM [22–24], the research context of the emerging Chinese market may lead to new insights in the GSCM literature and businesses.

The structure of this paper is as follows. In the next section, a thorough literature review on the GSCM strategic approaches and practices is provided. Based on the comprehensive literature review, we develop a conceptual framework for exploratory investigation

followed by the methodology of this study. After that, findings of an in-depth case study are presented followed by a synthesized discussion on the research findings. Finally, we summarize the conclusion and implications of the study.

## 2. Literature Review

GSCM practices cover a wide range of supply chain management activities that include process flows, relationship management from upstream sourcing and procurement, inbound production, and operations management; to outbound logistics, downstream transportation, packaging, and warehousing; close-loop reverse logistics, remanufacturing, and recycling [25–27]. Extant literature shows that different ways of GSCM can be used for businesses leveraging various sustainable competitive resources [2,6]. Table 1 summarizes two of the fundamental GSCM strategies. The associated applications, benefits, and shortcomings of the strategic taxonomy are analyzed in detail.

**Table 1.** Green supply chain management strategies.

| GSCM Strategies | Application | Benefit | Shortcoming | References |
|---|---|---|---|---|
| Pollution control and prevention | • Manufacturer adopting ISO 14000 series standards on production operation<br>• Manufacturer adopting environmental management systems with lean production system to improve standardized operations and reduce waste on high-production-volume and low-product-variety line | • Ensure regulation compliance, minimize negative impact on environment and risk of operational fine<br>• Detect internal waste of defects, scraps, inventories, reduce waste, energy consumption, emission<br>• Encourage material reuse and recycling | • Limited economic returns of adopting pollution control strategy that focuses on end-of-pipe process control and cut waste within each operational silo<br>• High cost of adopting a pollution prevention strategy that prevents the occurrence of pollution at the beginning of product development and waste recycling for total resource efficiency | [1,2,4–6,25–37] |
| Green product stewardships | • Manufacturer working with suppliers to design environmentally friendly products with consideration of environmental impact during product use and disposal | • Reduce liability and develop new products with lower life-cycle costs<br>• Improve positive impact of products on environment across lifecycle and stimulate novel market competitiveness | • Green value profit cannot always be capitalized due to unpredictable green market<br>• Undermine opportunities for competitive exploitation of advanced technology | [3,7–9,24,26,35,38–42] |

### 2.1. Green Supply Chain Strategies for Different Competitive Advantages

Pollution control and prevention strategies are widely studied for business to develop supply chain cost competitiveness [31,43–45]. Businesses adopting this approach see the

green supply chain as a way to reduce risks of violation in environmental compliance and reduce internal operational wastes. Early studies [30,46] focus on applications of ISO 14001 environmental standards to internal production. Manufacturers adopting ISO Environmental Management System (EMS) can standardize internal operational processes to meet compliance requirements and reduce negative impact of emission through end-of-pipe emission control and treatment. Principles of ISO 14001 are established on the Plan-Do-Check-Act (PDCA) cycle. They include a set of tools that help manufacturers identify out-of-control situations to avoid costly fines. The tools also help manufacturers identify the required resources to redesign procedures and processes for emission control, measure and monitor processes continuously with real-time pollution data, and keep them below regulatory limits. In the end, the tools help manufacturers report and review results of measurement to ensure compliance and green performance [5,47]. The focus of the strategy is on internal process improvement and minimization of negative impact on the environment due to pollution and risk of operational fines.

Applications of pollution control strategy are also widely discussed in areas of integration of EMS and lean production systems. Lean systems follow the similar concept of the PDCA continuous improvement cycle but focus more on identifying defects and scraps in the value streams and finding new ways to reuse them within the process. These studies discuss the high impact of lean and green control in identifying and controlling internal emissions. They also highlight the reduction in internal production wastes of defects, scraps, redundant inventories, and energy consumption using Min-Max level tools, especially on high production volume and low product variety lines [4,48,49]. Other studies [46,48,50] discuss the impact of lean and green control on packaging and delivery operations. They propose that, through adoption of ISO 14001 standards and lean approach to product packaging and delivery, businesses can identify ways of bulk packaging and centralizing delivery capacity to reduce total truck mileage and $CO_2$ emission.

Subsequent studies take a step further. Instead of focusing on end-of-pipeline control, they look at preventing the occurrence of pollution and waste in the first place. Rather than dealing with pollution concerns in a post-hoc manner, businesses can take a more active approach by going beyond the competitive outputs of green operation to exploit waste as a source of competitive advantage. Waste is redefined to include not only scraps but also restocked, refurbished, and recovered product units [51]. The environmental efforts are preplanned and coordinated into a deeper level of operations, such as substitution of nonrenewable materials by green sourcing, reduction in material use at product design, and extended operational integration for reserve and closed loop recycling operations [52,53]. Applications include ISO standards incorporating eco-design and end-of-life product safe disposal methods under compliance. Pollution prevention strategies adopted underpin a resource view for business leveraging an end-to-end cost cutting advantage. However, daunted by the higher price taken to conduct waste recycling, small- and medium-sized companies with constrained resources and technology find those strategies hard to adopt. Furthermore, the returned economic value is reported to be limited [12]. Compared to operating green processes and using recycled materials, traditional ways of operations and using raw materials are much cheaper to operate.

Green product stewardship strategies are considered as an advanced solution for businesses taking proactive environmental initiatives. Businesses adopting this approach see green methods as an opportunity to develop differentiated products and build market novelty through creation of sustainable products that can have less—or even zero—impact on the environment [7]. This requires development or adoption of emergent technologies or systems to modify traditional routines [54] and drive creative redesign of industries to embrace modern environmentalism [8]. Practical application of this approach involves working with suppliers to design more environmentally friendly products with consideration of environmental impact during product use and disposal [31,44]. Some companies design their products with consideration of life cycle environmental impact effects. For example, in the automotive industry, manufacturers are under increasing pressure to re-

duce emissions generated during the product-use stage as well as increase recyclability of components at the end-of-life stage [55].

Green product stewardship strategies leveraging technological advancements ahead of competitors support competitive pre-emption that can tap into novel market value [44]. For example, Brindley and Oxborrow (2014) [3] discuss the impact of organic material sourcing and collaborative menu design with local and specialty food suppliers to create a combination of food offerings and customized dishes to leverage a sustainable value-seeking market. Through more customized or personal exchange of information, collaborative adaptation of menus, postponement strategies, and intensive management of upstream sourcing activity through 'buying-to-order' (BTO), flexibility can be infused to improve value while overcoming sourcing uncertainties. However, businesses investing in eco-product innovation does not always translate to novel market value, especially in developing countries where environment infrastructure and institutional policies are not well built and awareness for green consumption is also not developed [24,56]. The market for green products is seldom viewed as lucrative at the initial stage. A few studies [57] point out that green product strategies focusing on clean innovation can undermine opportunities for competitive exploitation of technology advancements.

Latest studies [17,18] suggest that more context-specific solutions are needed to mitigate sustainability risk and improve market performance in supply chain sustainability practices to compete in modern day highly uncertain environments. As Laari et al., (2017) [58] assert, contemporary businesses aiming for competitive marketing differentiation need to pursue hybrid strategies of cost leadership and value differentiation with advanced GSCM strategies to manage their supply chains and take specific industrial contexts into consideration. Kumar and Rodrigues (2020) [4] also argue that supply chain characteristics need to be considered in GSCM adoption as a wider corporate strategy. They suggest that manufacturers with high production volume and low product variety should apply green and lean methods mostly in their production processes. The focus is on reducing waste generated from the production process and minimizing energy consumption using 5S condition of work area and standardized operations. In contrast, manufacturers with low production volume and high product variety should apply lean and green methods in product design and service delivery processes for just-in-time assembling and deliveries. The focus is to reduce the use of raw materials and the wastes generated.

### 2.2. Green Supply Chain Capabilities

A capability is the ability to do something. It can also be seen as an organizationally embedded non-transferable firm-specific resource whose purpose is to improve the productivity of other resources [59]. Traditional GSCM capability development focuses on accumulating resources through maintaining the market, controlling operational costs, reducing wastes, and optimizing the use of resources. Contemporary businesses focus on developing heterogeneity resources to innovate new green products, boost R&D for clean solutions, and develop new green markets [60].

#### 2.2.1. Green Operational Capabilities

A few studies in the operations and supply chain management literature state the importance of internal implementation and integration of practices as an antecedent to supply chain operations [61–63]. These studies suggest that implementation of internal green practices can lead to development of capabilities that complement or facilitate the extension of these operations to cross-functional supply chains [63,64]. Table 2 summarizes the key green operational capabilities for supply chain resource exploitation.

**Table 2.** Green supply chain operational capabilities.

| Capabilities | Related GSCM Practices | Outcomes/Benefits | References |
|---|---|---|---|
| Internal control, lean, and continuous improvement | • Increasing staff involvement on the shop floor and enhance familiarity with EMS and lean operations<br>• Achieving operational standardization in production, just in time distribution, and optimized inventory management | • Understanding internal problems better<br>• Achieving increasing operational improvement to reduce output of divisional wastes | [4,5,27,30,32,33,37,46, 47,50,61,65,66] |
| Cross-functional operation integration | • Adoption of EMS, lean across upstream supply chain flow to realize responsive ordering, demand-driven fulfillment, responsive reuse (including build-to-order supply chain)<br>• Step-by-step integration of downstream reverse flow including repair, reuse, and reverse logistics into forward processes | • Understanding problems holistically and improve extended knowledge sharing across the departments<br>• Turning wastes of one division into input of another across supply chain, reducing total input of material wastes | [4,24,30,41,50,67–71] |

Lean and continuous improvement initiatives are widely mentioned as a key internal capability in green operation literature [32,46,50,66]. Cost-competitive resources can be acquired through continuous improvement to existing functional operations and tradable resources across functions [5]. Dües et al., (2013) [30] propose that a common tool of the PDCA cycle in the ISO environmental system can help manufacturers achieve continuous improvement by increasing staff involvement on the shop floor and enhancing familiarity with EM operations. The concept of continuous improvement corresponds with the lean production value stream mapping system (i.e., six sigma and Kaizen tools). Through better quality control on sourcing, production, and disposal operations, manufacturers can understand problems holistically and improve extended knowledge sharing across the departments [37,66]. Kumar and Rodrigues (2020) [4] document that manufacturers who adopted ISO 14001 programs more often reap benefits from total quality management system improvement.

Cross-functional integration effort is also widely identified in the extant literature as a key capability to achieve closed-loop green operations [30,71]. Integration allows businesses to turn the wastes of one division into the input of another at supply chain flows across the divisions [69]. This capitalizes on the unused resource in a growth model and eventually pivots the traditional linear or separated thinking towards a circular economy. Sharma et al., (2010) [41] claim that manufacturers with an integrated sourcing, production and distribution system can deliver a responsive demand fulfillment that reduces the bullwhip effect of forecasting with accurate ordering and reduces redundant inventories at upstream. Sharma and Iyer (2012) [24] highlight the impact of reduction in reverse surplus through integration of product return, repair, and reuse operations into forward operational processes. Reverse supply chain integration enables businesses to carry out product use-life maintenance, end-of-life takeback, and repair with lower operating cost [72].

2.2.2. Green Relational Capabilities

Relationship management capabilities are widely discussed in the green supply chain and market management literature. They enable companies to go beyond boundaries

of individual business operations to share green resources proactively and explore new business value [5]. Table 3 summarizes the key green relational capabilities for supply chain value exploration.

**Table 3.** Green supply chain relational capabilities.

| Capabilities | Related GSCM Practices | Benefits | References |
|---|---|---|---|
| Collaboration with supply chain stakeholders | • Enhancing collaborative relationships with suppliers to develop green projects in areas of reducing pollution, reducing hazardous material input and eco-design<br>• Enhancing collaborative relationships with customers to improve product use life maintenance, repair, and reuse during service life<br>• Enhancing collaborative relationships with distributors to encourage applications of green logistics and develop product responsible takeback, refurbish, remarket, and recycle platforms | • Improving trust and openness of green information and knowledge sharing across industrial supply chain<br>• Enabling green and responsible development for industry growth | [1,3,15,31,35,39,73–80] |
| Dynamic collaboration and joint innovation | • Building new relationships with non-traditional stakeholder, including governmental bodies, policy makers, universities, and research institutes<br>• Improving in-depth learning and co-evolving across a group of partners for eco-design innovation | • Improving proactive and dynamic knowledge transfer with other industries<br>• Improving high quality information sharing, evaluation, and synergies across industries for green innovation | [6,17,20,22,23,38,40,70,81,82] |

Early GSCM practices have advocated the monitoring of suppliers' performances [77]. Bowen et al., (2001) [73] demonstrate that manufacturers can have more direct control on green compliance through monitoring suppliers in areas of pollution emission and controlling input of hazardous materials on component production.

Brindley and Oxborrow (2014) [3] suggest that involvement of suppliers in the early phase of design and purchasing can develop stronger knowledge sharing and mutual understanding of product development. Others extend this by arguing that more collaborative and in-depth relationships with supply chain partners are required for GSCM. Collaboration goes beyond the need of monitoring by a joint effort towards tackling environmental concerns that enables higher levels of improvement to be obtained throughout the supply chain [80]. It nurtures trust and openness on information-sharing across supply chain partners [35,39].

Similarly, Reche et al., (2020) [15] state that quality relationships with customers can enhance the understanding of consumer behavior, thereby allowing for personalized services. By working closely with customers, manufacturers can keep their customers' inventory at the desired levels and offer additional value-added services. Customers can buy a management service or the use of a product instead of purchasing real products that might place a burden on the environment. Collaboration among different stakeholders can bring different expertise and knowledge to the supply chain to create synergies across green

process and product practices. This is especially important in product life cycle assessment, requiring information sharing from all external organizations with the focal company.

Studies show that environmentally proactive firms appear to be more likely to form stronger collaborative relationships with their suppliers and customers. Green product stewardship strategies highlight a relational management view focusing on the development of extensive interaction with external stakeholders—such as suppliers, customers, regulators, communities, non-governmental organizations, and the media—for green collaboration and knowledge sharing [1,31]. Businesses can propagate green change through improving external supply chain stakeholder confidence in their intentions and activities, enhance corporate reputation and catalyze the spread of more sustainable practices within the business ecosystem [8].

More recently, Melander and Pazirandeh (2019) [40] argue that businesses need to go beyond traditional stakeholder collaboration (i.e., established suppliers, customers, and service providers) and work with other industrial entities to further enhance green innovation. New external stakeholders include governmental bodies, policy makers, universities, and research institutes. In-depth learning and co-evolving with various partners facilitate high quality information sharing, evaluation, and synergies which help businesses deliver proactive and context-specific solutions under high uncertainty [81,82]. Business partnerships that provide big data collection, data analytics, and diagnostics solutions can also foster dynamic knowledge sharing with new business partners [40]. To mitigate long-term challenges and create an organizational environment supportive of the innovation process, firms are required to proactively search for internal repositioning of skills with new knowledge and capabilities.

### 2.3. A Conceptual Framework of Competitive Green Supply Chain Transformation with Dynamic Green Capabilities

This study aims to explore the research question of how a company adopts green supply chain strategies for a competitive transformation. We take the view, based on theory of dynamic capability [83], that business develops difficult-to-replicate enterprise capabilities to adapt to changing environments through 'orchestrating' capacities for sustainable competitive advantage. Dynamic capability theory posits that—in fast-moving business environments open to global competition and characterized by dispersion in the geographical and organizational sources of innovation and manufacturing—attainment of sustainable advantage requires not only ownership of (knowledge) assets but also combination of multiple resources to achieve dynamic marketplace success [84,85]. Business innovation occurs through sensing the market, exploring technological opportunities, seizing opportunities through building strategic structures and incentives, and leveraging reconfiguration through combining assets [83].

To explore green supply chain transformation, we put forward a conceptual framework as shown in Figure 1. The proposed conceptual framework depicts the following: (i) green supply chain strategic positions; and (ii) green supply chain capabilities to achieve the associated positions. Strategic positions determine the direction for business green adoption. Green supply chain capabilities show how green initiatives can be integrated with supply chain management and product design for new competitiveness. As shown in the framework, the reviewed literature reveals a dichotomy of green supply chain strategies: (i) pollution control and prevention to minimize wastes in supply chain operations and reduce risk of violation in environmental compliance; and (ii) green product stewardship strategies to develop differentiated product value and improve environmental value in the supply chains.

**Leveraging dynamic green capabilities with combining choices**

**Sensing competitive opportunities and transforming green strategic focuses**

Green supply chain relational capabilities

- Supplier and customer collaboration
- Dynamic partners collaboration and joint innovation

Green supply chain operational capabilities

- Internal operational control, lean and continuous improvement
- Cross-functional operational integration

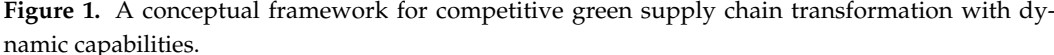

- To stimulate green product design and supply chain value
- To prevent pollution, reduce waste and improve supply chain resource efficiency

**Figure 1.** A conceptual framework for competitive green supply chain transformation with dynamic capabilities.

For the capabilities, the reviewed literature demonstrates that lean and continuous operational improvement within functions and cross-functional integration are the key capabilities for business to maintain a resource advantage and to push for competitive exploitation of supply chain sustainability. Comparatively, building collaborative relationships with supply chain stakeholders and new cross-industry partners with a high level of proactivity and joint efforts for eco-design are the key capabilities for business to create new resources and to pull from competitive exploration of supply chain sustainability.

In this study, we argue that through developing different dimensions of green supply chain capabilities across operational and relational levels, businesses can create dynamic capabilities to compete in a turbulent environment. These capabilities help businesses sense different external green opportunities and reconfigure supply chains for aligned green strategic positions across resource efficiency and value exploration. The framework serves as a guideline for us to observe—through an in-depth case study—the potential paths that a business can take to develop the right capability bundles to build resource dynamics for a specific green strategic transformation.

## 3. Methodology

The research question of this study is primarily exploratory. As we are investigating a newly emerging context, a qualitative research methodology is considered appropriate. A qualitative approach examines how individuals and groups develop an understanding of the social world [86]. We use the case study method involving in-depth and detailed examination of a particular case within the real-world context. This method enables researchers to deeply understand the intentions and motivations behind a unique and complex phenomenon [87,88].

GSCM adoption is still at an early stage in China. In 2016, China formulated the outline of an industrial green development plan (2016–2020) to highlight green manufacturing as a key direction of National Industrial Transformation [11]. The plan focused on developing: (i) green design platform; (ii) green operational skills and process upgrade; and (iii) supply chain restructuring for manufacturing system integration, taking EM into the wider SCM scope. The government targeted large manufacturers and used them as examples for green transformation of the entire industry. For in-depth investigation, this study selected one of the few exemplars of GSCM adoption, a multinational Information and Communications Technology (ICT) organization in China.

We conducted a longitudinal study to trace the development and transformation of the business under investigation in GSCM from 2009 to 2020. Longitudinal study helps

researchers access the richer information and observe the changes across time in a single setting [88]. We specify the analysis into four different product lines to observe patterns of variation across them in different phases of the green development, with a view to drawing some theoretical insights. The product categories include: (i) premium PC and laptop; (ii) tablets and smartphones; (iii) generic workstation and server; and (iv) fast consuming accessories.

### 3.1. Background of Case Company

As one of the largest personal computer vendors in the world, the case company operates in more than 60 countries (including high-end assembling production lines and R&D centers in the USA, Europe, and Japan) with its product selling to about 180 countries. According to the data of 2020, it had a market share of 25.7 percent of global PC shipments and was ranked the third largest computer maker in the world by volume.

The company's main product category is personal computers, accounting for about 80 percent of the company's revenue. Despite the strong growth in the global market, the company also owns a large domestic market accounting for over 40% of the market share. The company's corporate sustainability awareness came about through foreign business acquisitions and global market development. It started its EM transformation in 2005. To facilitate its adoption, the company set up a cross-functional EM team in parallel with business units in 2008 to support strategy adoption in different product categories. Since then, the company has worked on adjusting EM practices and reforming to align with local market conditions.

### 3.2. Data Collection and Analysis

This study utilizes multiple sources of information, including semi-structured interviews, on-site observations, reviews of business news, company reports, and databases for evidence and data triangulation. In-depth interviews were carried out with the key personnel responsible for EM, including the cross-functional EM team for product design, sourcing, production, and end-of-life operations of different product sets. We also conducted a focus group interview with the production manager and the operators at one of the large factories of the company. A few OEMs (original equipment manufacturers) of the company were also interviewed. In total, 12 semi-structured interviews with open-ended questions were carried out.

Thematic coding analysis was conducted using NVivo—a commonly used software for analyzing qualitative information. By matching the content with the descriptors, we developed the initial codes of concepts. A description of the coding categories is presented in Table 4. We used within-case unit analysis to identify supply chain patterns of green transformation in the different phases. Then, we compared patterns of similarities and differences across the case units to observe supply chain and market interactions across the development phases.

**Table 4.** Description of coding categories.

| High Order Categories | Example Codes | Example Citations |
|---|---|---|
| Internal operational control, lean production | ISO environmental management system, lean production, inventory and packaging waste control, recycle waste | " . . . most of the local factories in inland China have introduced quality production standards including ISO 90001 standard and lean and kanban assembling to reduce waste and cost of production . . . "<br>" . . . suppliers' components shall meet the substance restriction requirements of global hazardous substances regulations . . . "<br>" . . . we control not only the inventory of products but also components, and parts . . . we encourage suppliers to disclose the standardized number for the inventory of components . . . "<br>" . . . plastic packaging material is marked according to ISO 11469 referring to ISO 1043 . . . "<br>" . . . we use third-party recyclers to help us buy back used desktops from our major corporate users . . . this improves safe disposal while improving our reputation . . . "<br>" . . . from mass recycling of plastic materials, we only get a thin profit . . . " |

**Table 4.** *Cont.*

| High Order Categories | Example Codes | Example Citations |
|---|---|---|
| Agile and clean operational innovation | Green technology for zero emission operations, operation with renewable energy | " … in terms of green technology, we have introduced low-temperature solder paste technology. The temperature of the solder paste has a great influence on production reliability and improves clean operations and zero chemical emission at assembling … "<br>" … many production sites use solar panels to transform buildings and use more new energy to replace traditional energy … " |
| Cross-functional process integration | Build-to-order operation, just-in-time delivery, responsive takeback and repair, demand-driven order | " … we adopt build-to-order production and just-in-time responsive delivery downstream … "" … demand-led ordering enables better order accuracy and reduces over production, thus targeting parts inventory reduction at the beginning of production … "<br>" … end-of-life product redeem service is offered to consumers to encourage end-of-life takeback … "<br>" … consumers can choose from different choices and find matched ways for repairing or reselling products through our internal services … we also offer revaluation of products … " |
| Horizontal supply chain stakeholder collaboration for eco-design | Design to reduce, design for longer life, design for repair or material recycling with local partners | " … collaborate with suppliers to design products on modular standardization for easy assembling … "<br>" … encourage suppliers to reuse wasted industrial materials … "<br>" … work with local service points for repair and maintenance … upgrading can be done with processor, memory, cards and drives … "<br>" … use a single material or easily separable materials for plastic parts that are heavier than 100 g … " |
| Vertical collaboration for eco-design | Radical clean innovation with global industrial partners, dynamic collaboration across industries on green design projects | " … for core items, such as chips … we are not considering product easy disassembling as it goes against confidentiality agreement with some key suppliers."<br>" … introduce post-consumer recycled materials by purchasing from international suppliers onto the external enclosure of the server … "<br>" … work with international R&D factories and OEM suppliers on improving product power energy efficiency, portable configuration, smart charging, LED lighting and design for ergonomics … " |

## 4. Findings

### 4.1. Adoption of Green Supply Chain Agenda via Mergers and Acquisitions (M&A)

The company launched its EM agenda in 2005 when the company experienced a strong global market growth with a series of international brand acquisitions. During this stage of transformation, the business developed its general green production protocols based on global industry standards. It also initiated green product innovation to meet global market customer demand.

The products intended for the global market had to adhere to basic environmental standards of the industry, e.g., restriction in the use of hazardous substances (RoHS) for electronic products and baseline ISO 14001 system in areas of safe production and disposal operations. Compliance with these standards is compulsory for Chinese businesses to enter the European and the American markets. The company also introduced a series of new initiatives on innovation and developed collaboration with global suppliers and OEMs. These initiatives focused on green changes and product innovation targeting global high-end customer groups.

In the late 2010s, the company changed its business focus from global market expansion to local market development and supply chain cost reduction. During this stage of transformation, the company carried out a series of vertical process integration to reduce cost of operations, exploit existing resources within the system, and fulfill large unserved local demand. In 2015, the company expanded applications of product innovation into local product lines and exploited volume market green value of mass domestic consumers.

### 4.2. Stage I: Pollution Control and Green Product Innovation

During this stage, the company introduced baseline local EM standards (i.e., ISO 14001 certification) on sourcing and production processes to manage upstream production pollution. It focused on production and sourcing monitoring. The implementation of the standards had a big impact on generic product components (including workstations, servers, and

fast consuming accessories). Assembly of generic products tended to be stable, and given that components were relatively easy to source, the company could select suppliers from a variety of choices. To lower the cost, the company used to select ones that could provide the same quality but at lower prices. With the external green policies and standards to fulfill, the company had to raise the bar and reconsidered the cost to accommodate the quality. Generic products were also affected by the company's adoption of lean and ISO 9001 quality management systems. These products could be produced with standardization in large-scale assembly lines. Adopting lean methods helped reduce the cost of wastes through material scrap control and defective products repair, packaging savings, and inventory control. Downstream, volume buyback was also leveraged since most of these products were sold to domestic corporate users.

At the same time, to fulfill global market expansion goals and target niche markets, the company sourced from new globally based material suppliers that provided recyclable materials (e.g., post-consumer recycled plastics), and worked with OEMs to substitute hazardous materials with green materials. This was mostly practiced on the premium product lines. For example, it introduced post-consumer recycled plastics on premium desktops and laptops. Of total plastic parts' weighting more than 25 g, the use of recycled material content could be up to 15–20% for desktops and premium PC product covers and 10% for the external enclosures of the general server. In addition, the company also initiated a series of green design projects to boost product functionality. They included power efficient design, portable design, smart charging, LED lights for energy saving, and product ergonomic functionality. New partnerships were formed with global OEM factories and R&D centers, research institutions and agencies located in Japan and the USA through business acquisition. These design changes were made mostly on high-end products, including energy-efficient desktops, high-end laptops, and mobiles. Table 5 summarizes the key green initiatives that the business adopted during this stage of green transformation.

**Table 5.** Stage 1 GSCM transformation across four case units.

| GSCM Strategy | Internal Pollution Control and Lean | Radical Green Product Innovation |
|---|---|---|
| Generic workstation and server | Internal control, lean production, and scale recycle (++)<br>• Reselect suppliers based on ISO EM standard to fulfill basic production standards (e.g., factories located mostly in inland China with large scale presence)<br>• Lean impact of waste reduction on assembling line is significant due to economy of scale production<br>• Start to work with third-party service providers to conduct scale buyback and safe disposal-business users | Design to reduce, horizontal collaboration (+)<br>• Work with local suppliers to prevent hazardous material input such as batteries (e.g., RoSH standard) baseline and voluntary requirements<br>• Add over 10% of consumer recycled content to external enclosure of the server |
| Fast consuming accessories | Internal control, lean production, and scale recycle (++)<br>• Reselect suppliers based on ISO EM standard to improve quality (e.g., factories located mostly in inland China with large scale presence)<br>• Lean impact of waste reduction on assembling line is significant due to economy of scale production<br>• Start to work with third-party service providers to conduct scale buyback and safe disposal | No impact (∅) |

**Table 5.** *Cont.*

| GSCM Strategy | Internal Pollution Control and Lean | Radical Green Product Innovation |
|---|---|---|
| High-end desktop and laptop | Agile and clean production (++)<br>• Adopt automation to improve operational agility<br>• Technological innovation on production line to eliminate carbon emission<br>• Waste reduction is limited due to accommodation on quick reconfiguration and variety assembling<br>• Most suppliers (including international partners) had a rather good awareness on EM operations and are long guided by EM standard yet | Radical product eco-design, vertical collaboration (++)<br>• Collaborate with key suppliers to reduce hazardous material input on the product and add 25% post-consumer recycled plastic content in key components<br>• Use recycled materials in product packaging such as recycled fiber in corrugated packaging<br>• Work with international R&D factories and suppliers on power energy efficiency, portable design and smart charging, LED lights, ergonomics design<br>• Energy efficient products consume 25% less power than conventional models by using the most efficient components and better managing energy use when idle<br>• Improve post-consumer recycled content of rare earth elements in hard drives |
| >Mobile | Lean production (+)/agile and clean production (+)<br>• Adopt quick reconfiguration and variety assembling<br>• Adopt hybrid 'leagile' operation with high modularity component and JIT assembling to enhance mass customization efficiency | Radical product eco-design, vertical collaboration (++)<br>• Work with local suppliers to prevent hazardous material input in component design<br>• Work with local suppliers to use recyclable materials on packaging<br>• Work with international R&D factories and suppliers on power energy efficiency, portable design and smart charging, LED lights, ergonomics design |

++ represents this practice is being highly adopted, + represents this practice is being adopted to a limited degree, ∅ represents this practice is not being adopted.

*4.3. Stage II: Efficiency Improvement and Green Product Adaptation*

During this stage, the company identified a series of issues of high inventories and high cost of operations due to earlier global market expansion and product diversification. The company implemented vertical process integration to enhance operational efficiency. Upstream, the company enhanced standardization on high-end desktop, laptop, and mobile product design to reduce supply chain complexity resulting from over customization of products. Modular design was largely considered in these product categories. The company changed the traditional product ordering system to enable 'build-to-order' production and integrated distribution. As such, mass customization could be achieved at the high-end product lines, improving operational efficiency. It also reduced inventories and non-value-added activities in the supply chain. Downstream, process integration was also observed for high-end products. Information was shared between the service team and the logistics team to facilitate a responsive end-of-life product buyback, repair, and reuse. Parts that can be repaired were disassembled at point of sale and refilled to the upstream processes to improve internal stock and exploit resources fully within the system. It was especially highlighted on the premium fashionable products, i.e., mobile and tablets.

To minimize pollution and energy consumption, the company introduced low-temperature solder paste printing technology in surface mount assembly in 2017 to enable lead-free solder pastes to reduce carbon emission and conserve energy. These were adopted as pilot projects with the support of government policy. The company rolled out automation technologies in a few premium R&D factories in 2019. Machine automation transformed

the traditional production line with higher flexibility and reduced labor cost. In these factories, most workers still worked in traditional kanban-based production lines while being simultaneously trained to upgrade their skills required for job repurposing.

Meanwhile, to boost the domestic green market, the company adapted innovative green components to meet local emerging product market requirements. Relationships are reinforced between traditional suppliers and partners to share the cost and value of green product adaptation on the supply chains. For instance, design to reduce and recycle was considered on the high-volume generic products. The company considered the use fewer types of materials in generic component production to enable end-of-life recycling while maintaining functionality of the product. Design for repair was also considered. For example, alterations were made on the design specification to remove 25 g of plastic parts without using any chemical treatment. Use-life easy maintenance was also considered by the company for easy software upgrade and service support on generic workstations and servers to enhance customer value. These product-level changes helped the company further exploit local market resources while incrementally infusing green ideas into products. Table 6 summarizes the key actions on operation and product design at this stage of GSCM transformation.

**Table 6.** Stage 2 GSCM transformation across four case units.

| GSCM Strategy | External Operation Integration and Waste Reduction | Incremental Green Product Adaptation |
| --- | --- | --- |
| Generic workstation and server | Cross-functional operational process integration (+)<br>• Volume waste reduction in logistics<br>• Reduce surplus wastes through bulky packaging and delivery at warehouses<br>• Volume batch buyback and recycle on a regular basis | Design for reduce recycle, horizontal collaboration (++)<br>• Use less material type on product design followed by general guidance; for instance, about 25 g of plastic parts can be completely disassembled without any chemical means, only physical means<br>• Downstream disassembly and recycling are handled by authorized network<br>• Provide 5-year warranty service from the date of purchase; include key hardware maintenance and repair, and software upgrade<br>• Build long-term, close relationship with large corporate users who tend to buy in volume and use the product for a long time<br>• Easy tooling is considered to enable downstream general parts repair and maintenance |
| Fast consuming accessories | Cross-functional operational process integration (+)<br>• As the fast-consuming, low product value, and low takeback amount, reverse resource efficiency is not considered | Design for reduce recycle, horizontal collaboration (++)<br>• With high volume and fast perishable nature, simple design and cheap materials are considered for obtaining economy of scale production leading to some level of easy recycle |
| High-end desktop and laptop | Cross-functional operational process integration (++)<br>• Reduce total surplus wastes via BTO production and enhance order accuracy<br>• Improve efficiency of quick reconfiguration and delivery<br>• Takeback is encouraged with reduced cost and responsive redeem offering<br>• Introduce low-temperature solder pastes to reduce 35% of carbon emission and conserve energy with higher production stability<br>• Integrate automation in few R&D factories and work with ODMs in Japan and USA on skill improvement | Design for repair, horizontal collaboration (+)<br>• Modularity is considered for easy assembly on general parts, (e.g., chips and other technologically sensitive parts are integrated into the motherboard by the suppliers) |

**Table 6.** *Cont.*

| GSCM Strategy | External Operation Integration and Waste Reduction | Incremental Green Product Adaptation |
|---|---|---|
| Mobile | Cross-functional operational process integration (++)<br>• Reduce total surplus wastes via BTO production and enhance order accuracy<br>• Improve efficiency of quick reconfiguration and delivery<br>• Takeback is encouraged with reduced cost and responsive redeem offering<br>• Internal repair and resale become possible with lower cost | Design for repair, horizontal collaboration (+)<br>• Modularity is considered for easy assembly on general parts |

++ represents this practice is being highly adopted, + represents this practice is being adopted to a limited degree.

## 5. Discussions

The findings of the study show that both high-value supply chains (including personal desktop, laptop, and mobile products) and high-volume supply chains (including workstation, server, and accessory products) require a competitive 'hybrid' GSCM solution. In this section, we answer the research question of how a company greens its supply chain leveraging the dynamic capabilities of green supply chain operations, relationships, and product development for a competitive market transformation. We also demonstrate patterns of green supply chain and market transformation embedded in the different supply chains and summarize the key capability bundles (see Table 7 and Figure 2).

**Table 7.** GSCM transformation across different products leveraged by dynamic green capabilities.

| GSCM Strategies | Pollution Control and Prevention | | Green Product Stewardships | |
|---|---|---|---|---|
| Key capabilities | Stage I: Internal operational control and lean operations | Stage II: Cross-functional operational process integration | Stage I: Radical product eco-design | Stage II: Incremental green design |
| Generic workstation and server (high-volume) | Intra-operational control: Internal control, lean production, and scale recycle (++) | Inter-operational integration: Cross-functional operational process integration (+) | Horizontal collaboration: Design to reduce, horizontal collaboration (+) | Horizontal collaboration: Design for reduce recycle, horizontal collaboration (++) |
| Fast consuming accessory (high-volume) | Intra-operational control: Internal control, lean production, and scale recycle (++) | Inter-operational integration: Cross-functional operational process integration (+) | ∅ | Horizontal collaboration: Design for reduce recycle, horizontal collaboration (++) |
| High-end desktop and laptop (high-value) | Intra-operational innovation: Agile and clean production (++) | Inter-operational integration: Cross-functional operational process integration (++) | Vertical collaboration: Radical product eco-design, vertical collaboration (++) | Horizontal collaboration: Design for repair, horizontal collaboration (+) |
| Mobile (high-value) | Intra-operational innovation: Lean production (+)/agile and clean production (+) | Inter-operational integration: Cross-functional operational process integration (++) | Vertical collaboration: Radical product eco-design, vertical collaboration (++) | Horizontal collaboration: Design for repair, horizontal collaboration (+) |

++ represents this practice is being highly adopted, + represents this practice is being adopted to a limited degree, ∅ represents this practice is not being adopted.

Dynamic green capabilities

Strategic competitive focuses

| Intra-operational and horizontal relational capabilities combination |
| --- |
| • Internal EMS/Lean/Continuous improvement |
| • Supplier customer stakeholder collaboration |

High volume supply chain green transformation

→

| • To prevent pollution, reduce cost of waste |
| --- |
| • To exploit market eco-efficiency with operational responsiveness |

| Inter-operational and vertical relational capabilities combination |
| --- |
| • Joint innovation and eco-design |
| • Cross-functional integration |

High value supply chain green transformation

→

| • To stimulate green product demand |
| --- |
| • To explore market eco-differentiation with operational efficiency |

**Figure 2.** A revised framework of dynamic green supply chain transformation under competitive market shifts.

*5.1. Pollution, Waste Control, and Incremental Green Product Strategies on High-Volume Supply Chains*

High-volume supply chains adopted a green strategy of controlling production pollution and minimizing waste via lean production. These supply chains had implemented lean systems before the adoption of baseline ISO green standards. For bulky generic products, lean production systems facilitating standardized processes and waste control are effective in reducing defects and scraps on mass production lines. With standardized modular design and a stable demand, the systems are also good at preventing redundant component inventories by applying a just-in-time ordering system. Environmental management systems (EMS) helped high-volume supply chains detect unsafe code of conduct and find new ways to repair defects and reuse scraps on the production site using the PDCA cycle of improvement. We find that a high-volume supply chain is effective in adoption of pollution and waste control strategies since the benefits of economies of scale can be leveraged. Thus, waste is minimized; and resource efficiency of high-volume product production is improved [4,49].

Furthermore, EMS standards and lean systems were also applied to extended operations of high-volume supply chains, including downstream scale transportation, centralized reverse logistics planning, end-of-life product takeback, and recycling. Lean systems enable process standardization on bulk packaging and transportation, bulk buyback, and scale recycling thus reducing wastes in packaging, space, waiting time and enabling end-to-end resource recycling through continuous improvement of each functionality. EMS monitors processes, detects further carbon emissions, revises each process for end-of-pipe pollution control, and finds ways to repurpose wastes within these functions. Therefore, such green strategies enable high-volume supply chains to optimize internal operation and reduce waste in each supply chain function to gain an eco-efficient market advantage. These findings corroborate with that of previous studies [5,48], highlighting the impact of lean and green control strategies on the high-volume supply chain green transformation.

At the product level, high-volume supply chains underwent sequential changes to embrace green design practices. For instance, instead of redesigning products and modifying product structure radically, high-volume supply chains adapted the existing design and processes to phase out outdated practices that could be harmful to the environment. The company worked with local traditional suppliers to replace hazardous materials with safe materials that can have the same functionality while maintaining cost efficiency. Green

product improvements were also related to design for reduce-and-easy-recycling practices in terms of ability to use less materials and mono-material in the production of parts. All of these changes were made to existing product structure and supply chain process adaptations. They helped the company improve product quality and services while improving total supply chain operational efficiency. Thus, high-volume supply chains could further exploit product value, sustain existing customers and markets with enhanced services and resource efficiency. Interestingly, we find that through adopting gradual product shifts, high-volume supply chains can achieve an end-to-end supply chain resource conservation with reducing cost advantage.

Previous studies highlight the link of radical green product strategies with high-value product ranges for boosting value-seeking market differentiation [3,44]. We also find that a high-volume supply chain can exploit supply chain and existing market value through adoption of waste control strategy and incremental green product strategy. These strategies help firms adapt product design and associated supply chain processes in steps while fully exploiting current supply chain resources and value of mass market green development.

Integration of Intra-Operational and Horizontal Relational Capabilities

The findings show that a mix of green operational and relationship capabilities were used to facilitate the high-volume supply chain green transformation. We find that high-volume supply chains focus on leveraging business internal capabilities to reduce the negative impact of operations on the environment. For instance, the company developed continuous operational improvement under adoption of lean production and standards of ISO 14001 and 9001 systems with associated tools including the PDCA cycle. In previous research, Dües et al., (2013) [30] propose that the PDCA cycle as a common tool in the ISO environmental system can help manufacturers achieve continuous improvement by increasing staff involvement on the shop floor and enhancing familiarity with EM operations. The impact of lean and continuous improvement on internal operational waste reduction, especially on high-volume products, is documented in literature [33,66].

We find that rather than only focusing on building internal operational capabilities for pollution and waste control, external relationship management is also highlighted in the findings for high-volume green supply chain transformation. These include practices of monitoring external suppliers and stakeholders to fulfill environmental compliance. Key traditional suppliers are monitored for green baseline compliance. The findings support that of previous studies [77,78] on the importance of supplier monitoring for green changes. The company encourages its traditional component suppliers to adopt lean and green systems to ensure a baseline component production control on generic products.

Furthermore, the findings also suggest that high-volume supply chains can improve compliance-based relationships established on monitoring and controlling to a more collaborative partnership through working on green projects and initiatives. These projects are mutually beneficial since they also help traditional small and medium-sized suppliers to pass compliance, reduce their risks of operations, and improve their operational resource efficiency in the long run. High-volume product manufacturers thus leverage horizontal relational collaboration on the supply chains through maintaining and reinforcing relationships with the existing key suppliers and customers. Rather than looking for new green stakeholders, manufacturers can fulfill green baseline compliance and reduce risks of supply chain restructuring.

By building trust with a group of selected stakeholders including suppliers and recyclers, the cost of adopting a pollution control strategy can be shared and a high level of green commitment can be better communicated. Upstream, collaboration with a group of key partners enables lean businesses to consolidate supplier bases into accredited sources, such as large established wholesalers, who can meet sustainability objectives while still maintaining a cost advantage through pushed capacity [3,74]. Downstream, by reinforcing relationships with long-term key customers, lean businesses can enhance green service value while delivering bulk buyback at end-of-life of product disposal [51]. Through

building collaborative relationships with a few traditional supply chain stakeholders horizontally, a centralized green network can be developed for pushing operational capacity with an economy-of-scale advantage while balancing the cost of green operational change.

*5.2. Radical Green Product Innovation and Waste Prevention Strategies on High-Value Supply Chains*

A high-value product supply chain initiates green transformation by adopting radical green product innovation. The company initiated a series of green technological innovations for its premium product lines. This included improving the functional level of energy saving batteries; LED light switching; material level of introducing recyclable and durable materials, such as plastics; and structure level of design for easy disassembling with high modularity. Product change, by utilizing more green components, aimed to build a positive environmental image of its products that helped create a first-mover competitive market. In line with product innovation, artificial intelligence and clean production technologies were also applied on the assembling line. The adoption of these technologies—such as robotics and low-temperature solder paste technologies—helped the company to achieve a new level of precision and flexibility in monitoring tasks, controlling pollution, protecting the human workforce from hazardous working environment, and minimizing operational inconsistency. It also drove the high-end product lines to accelerate production automation process and transformed to an integrated manufacturing process system. These findings corroborate with that of Bhupendra and Sangle (2016) [28] and Sharma et al., (2010) [41] that businesses adopting green product stewardship strategies integrate green into product life-cycle management to proactively design green products and drive novel business value. Our findings support previous studies [3,44] in that low-production-volume and high-value supply chains are more suitable to adopt radical clean innovation to build first-mover market effects and stimulate novel market value.

In addition, it was interesting to note that our findings also suggest that high-value product supply chains can adopt operational waste prevention strategies to maintain high-value green market growth. To fully exploit resource advantages led by technological innovation, waste prevention initiatives help high-value supply chains, which were used to operate under agile operations, reduce non-value-added activities and operational cost, and enhance efficiency across the supply chains. Thus, high-value supply chains can improve operational variety efficiency in delivering to differentiated markets, thereby improving the total resilience of the operations. The applications include adoption of mass customization at production, just-in-time delivery systems, and built-to-order practices to improve integrated operations, and reduce over variation in production, ordering, and inventory management while fulfilling differentiated demand changes. Design for repair and reuse including component modularity on high-value products can be considered with the adoption of a waste prevention strategy to fully exploit unused value of the product. This is also achieved by working collaboratively across upstream and downstream supply chain partners including logistics service providers and customers for extended servicing and maintenance, collection points for damaged devices takeback, and repair service providers for repair or refurbish to realize an end-to-end waste stream reduction and resource conservation. The findings add to that of previous studies [52,53] on the linkage of high-value supply chains with the pollution prevention strategies.

Integration of Inter-Operational and Vertical Relational Capabilities

The findings show that a mix of green inter-operational and relational integration capabilities can be adopted to facilitate the high-value supply chain green transformation. In the case study, the company proactively sought new partnerships with global suppliers and OEMs and set up overseas offices and factories close to target global markets to boost supply chain innovation. The collaboration went beyond established stakeholders—including upstream traditional material suppliers, logistics service providers, downstream repair and refurbish service providers—to new green material suppliers, OEMs, and partners from other different industries including global research entities and universities on

joint clean technological and product design initiatives. This finding supports previous studies [38,40,81,82] in that environmentally friendly entrepreneurial (or enviropreneurial in short) product innovation requires dynamic collaboration, in-depth learning, and information sharing with specialized partners.

Our findings further identify that inter-operational green capability is also necessary for high-value supply chain green operation transformation. In the case study, high-value products are delivered by a 'build-to-order' mode of integrated operations. This helps the company enhance quick reconfiguration and responsive delivery while keeping the cost in control with certain operational standardization. Sharma et al., (2010) [41] contend that manufacturers with an integrated sourcing, production, and distribution system can deliver a responsive demand fulfillment that reduces the bullwhip effect of forecasting with more accurate ordering and reduces redundant inventories at upstream flow. Agile operation and process integration enabling higher supply chain flexibility also helps businesses develop effective reverse supply chains with responsive product returns, repair, and reuse operation [24] (Sharma and Iyer, 2012). Thus, we find that building cross-functional integration on the supply chain and leveraging inter-operational capability is more effective on greening high-value supply chains. Through developing cross-functional integration, a high-value supply chain can reduce non-value-added activities, turn the wastes of one division into the input of another at supply chain flows across the divisions to improve resource efficiency while exploiting green product value across the system. In this way, total supply chain resilience is also enhanced for high-value businesses competing in a highly unpredictable market.

## 6. Conclusions

The case study conducted across different product categories has illustrated how the case company greened its different supply chains at the operational process and supply chain levels to leverage a dynamic competitive eco-market transformation. The case findings show that high-volume lean supply chains leverage a hybrid of pollution control strategies and incremental green product strategies. This enables high-volume supply chains to enhance green compliance while maintaining cost advantage, improving total resource efficiency, and sustaining the existing eco-efficient market with enhanced services and products. To facilitate the strategic shift, high-volume supply chains focus on building internal green capabilities including lean and green and continuous process improvement to reduce waste streams within operational function. They also aim at enhancing existing supply chain stakeholder relationships horizontally to enable incremental changes in product design and supply chains for compliance and resource conservation.

The findings also indicate that high-value supply chains leverage a hybrid strategy of radical green production innovation and pollution prevention strategies. This enables high-value supply chains to stimulate first mover green product market for premium value while maintaining system agility for a differentiated market delivery. Such strategy also improves supply chain resource efficiency for market variety expansion. To facilitate this strategic shift, high-value supply chains focus on building dynamic vertical supply chain partnerships across industries to enable dynamic knowledge sharing, stimulate innovation, and cross-functional supply chain process integration for improved operational flexibility and system efficiency.

This exploratory study is among the few studies that discuss the dynamic GSCM strategic transformation from operation, relationship, and design levels highlighting the link of GSCM and green market management. The findings highlight the qualitative links of greening traditional high-volume supply chains for eco-efficient markets, and greening high-value supply chains for eco-differentiation markets [3,4]. The findings further extend the rigid GSCM strategies by proposing a 'hybrid' solution across supply chain operation, relationship management, and product design levels for dynamic green market transformation. Specifically, evolving patterns across two basic strategies and associated capabilities are demonstrated. The result illustrates that—rather than a once-and-for-all

adoption—green supply chain strategic transformation should be aligned with associated upcoming market challenges for a resilient operation and product adjustment across internal and external supply chains. In summary, we propose the following managerial implications for business GSCM adoption considering the nature of different supply chains.

- For high-volume supply chain green transformation, businesses can start with internal waste and operational control through the adoption of EMS and lean production system, and continuous process improvement to increase operational efficiency within each supply chain function.
- For high-volume supply chain green transformation, businesses can consider reassessing and reinforcing relationships with existing suppliers and customers to infuse operational responsiveness and integrated green collaboration in the process of product design and service management and improve total supply chain resource efficiency.
- For high-value supply chain green transformation, businesses can consider adopting green innovation on product design and operations. This can be achieved by building vertical collaboration and knowledge sharing with a dynamic group of industrial partners to stimulate green product design and market novelty.
- For high-value supply chain green transformation, businesses can further enhance process standardization on green design and product differentiation. They can build internal supply chain process integration across different functional departments to infuse operational efficiency for variety market expansion.

This study has a few limitations. Firstly, the result of the case study findings is generated based on a single case study despite the case company having a significant market presence and industrial reputation and is an exemplar in GSCM in China. We believe that theoretical replications of this research could be improved by exploring a wider group of case samples in future to observe any variation in other business settings. Secondly, the findings of this research are generated by studying the electronics industry. Although this industry is regarded as a representative context for GSCM studies, the industrial background of EM can differ greatly across industries. Future studies can also explore different paths of green transformation across industries to help improve the generalizability of the findings.

**Author Contributions:** Conceptualization, Y.Y. and K.H.L.; investigation, Y.Y.; data analysis, Y.Y.; validation, K.H.L.; writing—original draft preparation, Y.Y.; writing—review and editing, Y.Y. and K.H.L.; supervision, K.H.L.; project administration, Y.Y.; funding acquisition, Y.Y. All authors have read and agreed to the published version of the manuscript.

**Funding:** This research and the APC was funded by Academic start-up grant for young scholars of Soochow University, grant number NH10200822.

**Institutional Review Board Statement:** Not applicable.

**Informed Consent Statement:** Not applicable.

**Data Availability Statement:** Not applicable.

**Acknowledgments:** We would like to thank the company and associated personnel especially the project managers of ICT product lifecycle assessment and data service platform who provided us valuable information for this research; and we thank Soochow University for offering the funding for this study.

**Conflicts of Interest:** The authors declare no conflict of interest.

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
