# Peer review of "Competitive Green Supply Chain Transformation with Dynamic Capabilities—An Exploratory Case Study of Chinese Electronics Industry"

_sustainability, doi:10.3390/su14148640_

Round 1
Reviewer 1 Report
This paper is a resubmission and in the previous review of this paper, I have proposed to reject it. After improving it, I can't see enough modification to can ensure adequate quality. Unfortunately, my opinion is that paper hasn't enough quality to be published in Sustainability.
Author Response
Thank you for your comments.
In this revision, we have continuously reinforced the introduction, the discussions and conclusions chapters majorly. Now the research aim and question are specifically highlighted in the introduction chapter. In the discussion and conclusion chapters, we provided more elaborations on the key finding results established on the rising themes. The themes generated from literature review and newly rising themes generated from the findings were carefully compared and elaborated to highlight the key contribution of this study. The alignment of introduction, the research aims and questions, and findings of the whole has been improved.
Please refer to all the track-changes in the manuscript and the table 7 of the study. The manuscript was also proofread thoroughly with refined language editing. All the references were double checked to ensure the consistency and clarity.

Reviewer 2 Report
Comment: The introduction should be explained constructively, but not in so much longer writing. The research question is not established in the introduction section. It should be taken care of. It is not solved yet. See these papers (Reduction of waste and carbon emission through the selection of items with cross-price elasticity of demand to form a sustainable supply chain with preservation technology; Ramification of remanufacturing in a sustainable three-echelon closed-loop supply chain management for returnable products; Joint pricing and inventory model for deteriorating items with maximum lifetime and controllable carbon emissions under permissible delay in payments; Optimum sustainable inventory management with backorder and deterioration under controllable carbon emissions; Autonomation policy to control work-in-process inventory in a smart production system) and write that way with the comparative studies for green logistics issues.
Comment: The literature must be maintained sequentially for years. The author contribution table should be formed to show the model's novelty. See that paper, make an author’s contribution table, and compare it with all referred references in the reference section.
Comment: Write proper managerial insights to show the industry managers' benefit from this research. See that paper and write it that way.
Comment: Please write the significant findings in conclusions. Do not mention all assumptions which have been indicated within the model. See that paper and write it that way.
Author Response
Thank you for your pertinent comments. In this revision, we have continuously reinforced the introduction, the discussions and conclusions chapters majorly. Now the research aim and question are specifically highlighted in the introduction chapter.
Comment 1:
We have revised the introduction to emphasize the key content. Research questions were stressed in the article and further specified with sub-questions for clarity. The suggested articles are carefully reviewed and added into the literature review of this research. Please refer to introduction of the manuscript, Table 1-3 and reference list for the relevant changes.
Comment 2:
This study adopts a more holistic approach to carry out literature review. Instead of developing review based on the historical sequence of the previous literature, we focus more on identifying the theoretical links of existent studies. The taxonomies used to categorize the streams of studies were based on systematic review of the established GSCM theories and practices. They are also greatly aligned with our research aim and questions to highlight the existing theoretical structure and qualitative relationships across the sub-themes. Through this structural literature review process, the research gap and the study niche thus can be explicitly demonstrated with high research consistency.
Comment 3:
We have taken your advice to further synthesize the key findings. In the discussion and conclusion chapters, we provided more elaborations on the key results established on the rising themes. The themes generated from literature review and newly rising themes generated from the findings were carefully compared and elaborated to highlight the key contribution of this study. The alignment of introduction, the research aims and questions, and findings of the whole has been improved.
Please refer to discussion chapter in the manuscript and the table 7 of the study for the relevant changes.
Comment 4:
In the conclusions chapter, we have taken the suggestions to reinforce the key implications of this study and specific the limitations and contributions of this study. Please refer to the conclusion chapter in the manuscript.

Reviewer 3 Report
The paper has been improved, I agree to publish it in its present form.
Author Response
We appreciate your comment. The manuscript was continuously refined and proofread thoroughly with extensive language editing. All the references were double checked to ensure the consistency and clarity. Please refer to all the track-changes in the manuscript and table file.
Reviewer 4 Report
I am pleased to note that the authors have remedied the shortcomings I pointed out in my previous review.
Therefore, I consider that the article can be published.
Author Response

(The authors gave the same response as above.)

Round 2
Reviewer 1 Report
My opinion is still that the paper needs a revision, but based on the opinions of other reviewers and some corrections made by authors I will give minor corrections.
You should add some contributions in the introduction and should add a novelty described also.
Add these two references in literature review:
Kazemitash, N., Fazlollahtabar, H., & Abbaspour, M. (2021). Rough best-worst method for supplier selection in biofuel companies based on green criteria. Operational Research in Engineering Sciences: Theory and Applications, 4(2), 1-12.
Durmić, E., Stević, Ž., Chatterjee, P., Vasiljević, M., & Tomašević, M. (2020). Sustainable supplier selection using combined FUCOM–Rough SAW model. Reports in mechanical engineering, 1(1), 34-43.
Figures 1 and 2 should be more quality.
Ensure limitations and managerial implications.
Future research should be extended.
Author Response
Thank you for your comments.
We have revised the introduction, discussion findings and conclusions to further highlight the contribution. We also improve the parts of limitations, managerial implications and future research for the conclusion chapter. The recommended references have been all added to the study now.
Please refer to the highlighting changes of the manuscript and table files.

Reviewer 2 Report
I cannot get the final paper. However, I exactly want to see the comparison study to check the novelty. The paper can be accepted if this table with the comparisons will be inserted in the paper properly. Because it contains the novelty of the model.
Author Response
Thank you for your comments.
We have revised the introduction, finding discussions and conclusions to highlight the novelty of the study. Comparative analysis was highlighted in this version to show the contribution of findings with previous studies.
Thematic coding analysis shown in the table 7 was revised to clarify the key codes generated from findings. We also improved the articulation of comparative analysis of the findings based on this table.
Please see the highlighting areas of the revised manuscript and table files.
This manuscript is a resubmission of an earlier submission. The following is a list of the peer review reports and author responses from that submission.
Round 1
Reviewer 1 Report
The paper Greening supply chain process and product with dynamic capabilities – an exploratory case study of Chinese electronics industry falls within the scope of the journal, but according to my opinion, has not enough quality to be accepted and published in a journal such as Sustainability. My decision is to reject for the following reasons:
Clearly described aims, the main contributions, novelty, and verification of results are missing in the paper. The overall structure of the paper and quality is very poor.
What new brings your paper? What is your novelty, advantage?
Your qualitative analysis does not convince me that your paper deserves attention.
Any clear and concise methodology hasn't been applied.
I don't see the contribution of your paper to the wider community.
Reviewer 2 Report
The paper has several major issues regarding novelties, writings, and contributions. I have the following comments as follows:
- The format of the paper is a big issue. Correct the format in a proper way. The way of abstract writing is not perfect, and the abstract should contain the details of the study and the findings in a very constructive way.
- The research gap should be adequately explained. In the introduction, please rearrange/rewrite so that each authors’/most of the authors' contributions should be linked. Please try to maintain the literature sequentially. The comparative study with these papers, “A supply chain model with service level constraints and strategies under uncertainty; The Selection of the Sustainable Suppliers by the Development of a Decision Support Framework Based on Analytical Hierarchical Process and Fuzzy Inference System” theoretically is needed in the last part of the introduction to show the novelty of this study.
- The introduction should be based on the exact research gap, and the literature review should be based on the specific keywords-based review, and finally, make an author's contribution table to show the novelty and effectiveness of the study. Show all referenced papers in the table to show the contribution of this study.
- Write proper managerial insights to show the industry managers' benefit from this research and compare this study with “A new framework for the sustainable development goals of Saudi Arabia; Synergic effect of reworking for imperfect quality items with the integration of multi-period delay-in-payment and partial backordering in global supply chains" theoretically and methodologically the applicability of the proposed research.
- Please write the significant findings in conclusions. Do not mention all assumptions which have been indicated within the model.
- What is the data source of the numerical experiment? Please mention that the data is from industry or literature, i.e., accurate data or artificial data.
Reviewer 3 Report
The article is of interest, it is well structured, the research is well argued, the results are well explained, but we did not find the working hypotheses and their demonstration in accordance with the subject of the study.
Reviewer 4 Report
I very much enjoyed evaluating the article titled "Greening supply chain process and product with dynamic capabilities– an exploratory case study of Chinese electronics industry"
I believe that the article can be published once a few minor bugs have been worked out:
References have to be made in the format established by MDPI.
In the introduction the authors should clearly state the main contributions of the article.
The structure of the article should be revised and adapted to the standards established by the journal.